# Development and Field Application of a Diffusive Gradients in Thin-Films Passive Sampler for Monitoring Three Polycyclic Aromatic Hydrocarbon Derivatives and One Polycyclic Aromatic Hydrocarbon in Waters

**Shiyu Ren, Liangshen Li, Yucheng Li, Juan Wu and Yueqin Dou \***

School of Resources and Environmental Engineering, Anhui University, Hefei 230601, China;
renshiyu1998@163.com (S.R.); x23201038@stu.ahu.edu.cn (L.L.); li-yucheng@163.com (Y.L.);
wujuan@ustc.edu (J.W.)
\* Correspondence: yqindou@163.com

**Abstract:** Polycyclic aromatic hydrocarbon (PAH) derivatives are widely present in the environment, and some are more hazardous than their parent PAHs. However, compared to PAHs, PAH derivatives are less studied due to challenges in monitoring as a result of their low concentrations in environmental matrixes. Here, we developed a new passive sampler based on diffusive gradients in thin films (DGT) to monitor PAH derivatives and PAHs in waters. In the laboratory study, the XAD18-DGT device exhibited high adsorption rates and was demonstrated to be suitable for deployment in environmental waters on the timescale of months. The diffusion coefficients, $D$, were $5.30 \times 10^{-6}$ cm$^2$ s$^{-1}$, $4.51 \times 10^{-6}$ cm$^2$ s$^{-1}$, $4.03 \times 10^{-6}$ cm$^2$ s$^{-1}$ and $3.34 \times 10^{-6}$ cm$^2$ s$^{-1}$ for 9-fluorenone (9-FL), 1-chloroanthraquinone (1-CLAQ), 9-nitroanthracene (9-NA) and phenanthrene (Phe), respectively, at 25 °C. The DGT device's performance was independent of pH, ionic strength, deployment time and storage time, indicating it can be widely used in natural waters. In the field study, the target pollutant concentrations measured by the DGT are in good accordance with those determined via grab sampling. Then, the DGT devices were utilized to quantify PAH derivatives and PAHs in several rivers in Hefei, China. This work demonstrates the feasibility of using the DGT technique to detect trace PAH derivatives and PAHs in waters.

**Keywords:** PAH derivatives; XAD18-DGT; adsorption; field application

## 1. Introduction

Polycyclic aromatic hydrocarbons (PAHs) are a group of persistent organic compounds that are widely found in various types of environmental matrixes and have teratogenic, carcinogenic and mutagenic properties [1]. PAHs can be chemically and microbially transformed into PAH derivatives, including nitro polycyclic aromatic hydrocarbons (NPAHs), oxygenated polycyclic aromatic hydrocarbons (OPAHs), chlorinated polycyclic aromatic hydrocarbons (ClPAHs), etc. [2,3]. Certain PAH derivatives are more biotoxic than their parent PAHs [4,5]. NPAHs and OPAHs have a higher water solubility, and they are theoretically more mobile in water than their parent PAHs [6]. Compared to pyrene, the NPAH 1-nitropyrene exhibits a higher carcinogenicity and mutagenicity/genotoxicity in in vitro and in vivo tests [7,8]. A study in Brazil showed the mutagenicity of OPAHs in particulate matter samples increased approximately fourfold compared to their corresponding PAHs [9,10]. ClPAHs, which are structurally similar to dioxins and polychlorinated biphenyls (PCBs), are a type of highly toxic organochlorine compound [11–13].

Accurate and stable determination of compounds in an aqueous environment can help us better understand their transport and transformation processes to assess their biological toxicity and impact on the ecosystem and human health [14]. At present, in water, PAH derivatives and PAHs are mainly detected via spot grab sampling, demanding a significant amount of resources and expertise. Moreover, some trace-level pollutants

are often undetectable, or their measurements are susceptible to detection errors, which is not conducive to understanding the current pollution status and pollution trends in water [15,16]. Spot grab sampling provides instantaneous concentrations [17], which may not be representative of the pollution over a period of time or describe the temporal variations in the water quality [18,19].

Passive sampling techniques, requiring no collection of water samples or their transport and preservation, can save considerable human, material and financial resources compared with grab sampling. Furthermore, the time-weighted average (TWA) concentration measured by passive sampling is more meaningful, determining the overall water pollution during the assessed period [20]. Some studies have utilized passive samplers, such as semipermeable membrane devices (SPMDs) and polyethylene (PE) passive samplers, to quantify PAHs in water [21,22]. The SPMD procedure is well established and widely used. However, the presence of glyceryl trioleate in the SPMD makes the postanalytical purification process cumbersome [23]. The small capacity of PE passive samplers makes them unsuitable for long-term water monitoring [24,25]. The diffusive gradients in thin films (DGT) technique was first proposed by Davison and Zhang and was originally used to detect trace metals in freshwater systems [26,27]. Compared with other passive samplers, DGT passive sampling is less affected by environmental conditions, and laboratory-measured diffusion coefficients could be used to accurately predict in situ sampling rates, which would be highly convenient for field sampling. DGT techniques have also been widely used for trace organic contaminant detection in water [16,28]. However, to date, there have been limited studies on DGT or other passive sampling techniques specifically designed for sampling PAH derivatives in water.

In this study, we developed a new DGT passive sampler for monitoring PAH derivatives and PAHs in waters. Binding gels based on the XAD18 resin can rapidly adsorb a variety of organic pollutants in water and are simple and inexpensive to produce [29]. Thus, we used XAD18 binding gels as the adsorption phase and agarose gels as the diffusion layer material, which are resistant to temperature, pH and ionic strength changes [17]. The aims of this study are (i) to develop a DGT passive sampler for sampling PAH derivatives and PAHs in waters and study their adsorption kinetics, adsorption capacity, diffusion coefficients D and environmental effects in laboratory tests; (ii) to evaluate the DGT technique in field tests; and (iii) to apply the DGT technique for in situ monitoring of PAH derivatives and PAHs in rivers and lakes.

## 2. Materials and Methods

### 2.1. Chemicals and Analysis

In this test, we selected three PAH derivatives—9-fluorenone (9-FL), 9-nitroanthracene (9-NA) and 1-chloroanthraquinone (1-ClAQ)—which are representative OPAHs, NPAHs and ClPAHs, respectively, and one typical PAH, phenanthrene (Phe), as the target compounds. Their physical and chemical properties are shown in Table S1 in the Supporting Information (SI). The composition and sources of the other reagents and materials are shown in SI Table S2.

### 2.2. DGT Preparation

A standard DGT device consists of a filter membrane, an agarose-based diffusive gel and a binding gel. A detailed description of DGT device construction and its principles is presented in the SI. In this experiment, 0.8 mm diffusion agarose diffusion gels were prepared according to reference [30]. In order to select DGT-appropriate materials that have a low influence on the adsorption of the target compounds, agarose diffusive gel discs and four types of filter membrane (polyethersulfone (PES, 0.14 mm thick), polytetrafluoroethylene (PTFE, 0.18 mm thick), polyvinylidene fluoride (PVDF, 0.11 mm thick) and nylon (0.11 mm thick) (all with 2.5 cm diameter)) were separately immersed in a 100 mL solution containing the four target compounds (9-FL; 1-CLAQ; Phe: 100 µg L$^{-1}$; 9-NA: 20 µg L$^{-1}$). At the same time, DGT moldings were immersed in 200 mL of a solution of the same

composition. All of the above solutions were shaken for 12 h, and the adsorption of the four compounds on each material was calculated based on the concentration difference from the beginning to the end of the experiment [30]. The reason for the different concentrations of the target compounds is the low solubility of 9-NA and the fact that the 9-NA concentration is usually lower than that of the other three compounds in ambient water. The selected concentrations of each compound in the following experiments are the same as in this section due to the same reason.

The binding gel preparation procedure was as follows: The XAD18 resin was dried and ground into powder form, fully activated with methanol and then rinsed with ultrapure water. Then, 6 g (wet weight after pretreatment) of resin was added to 30 mL of 2% agarose solution, and then the solution was heated to boiling. This agarose solution was pipetted between two preheated glass plates (>80 °C) separated by a 0.5 mm thick PVC plastic sheet and cooled to room temperature. Then, the gel was cut into discs 25 mm in diameter. Diffusive gels and binding gels were placed in ultrapure water, sealed and stored at 4 °C.

### 2.3. Characteristics of the Binding Gel

### 2.3.1. Adsorption Kinetics

The ability of the XAD18 binding gel to rapidly adsorb compounds is key to its ability as a binding layer in DGT devices, and thus its adsorption kinetics need to be examined. The XAD18 binding gel was immersed in 100 mL of a solution containing the four target compounds (9-FL; 1-CLAQ; Phe, 100 μg L$^{-1}$; 9-NA, 20 μg L$^{-1}$), and shaken for 30 h. Samples were taken from the solution at different time intervals ranging from 5 min to 30 h for analysis. The mass of the target compounds adsorbed by the binding gel at different time intervals was calculated.

### 2.3.2. Elution Efficiency

The binding gels were immersed separately in 50 mL solutions containing the four target compounds at concentrations of 10 μg L$^{-1}$, 20 μg L$^{-1}$ and 100 μg L$^{-1}$ and then shaken for 4 h. After the adsorption was complete, the binding gels were placed into 10 mL of methanol and sonicated for 90 min; these solutions were then filtered through an organic filter membrane and analyzed via HPLC. The elution efficiency was the ratio of the measured mass of the target compound in the eluate to the adsorbed mass calculated by the difference in concentrations of the solution before and after adsorption.

### 2.3.3. Adsorption Isotherm

Adsorption isotherm tests were conducted to assess the uptake capacity of the binding gels. Different concentration solutions were prepared; the initial concentrations of 9-FL, 1-CLAQ and Phe were 100 μg L$^{-1}$, 500 μg L$^{-1}$, 1 mg L$^{-1}$ and 2 mg L$^{-1}$, and the initial 9-NA concentrations were 50 μg L$^{-1}$, 100 μg L$^{-1}$, 200 μg L$^{-1}$ and 300 μg L$^{-1}$. Next, the ¼ XAD18 binding gels were immersed into the solution of each component separately. The resulting concentrations of the solutions and the mass of the four target compounds loaded on the binding gels were measured after shaking the solutions for 24 h.

### 2.4. Diffusion Coefficient Determination

The concentrations determined via the DGT technique were calculated using Equation (1) [31].

$$C_{DGT} = \frac{M \Delta g}{D A t}, \tag{1}$$

where $M$ is the mass accumulated on the XAD18 binding gels, $\Delta g$ is the thickness of the diffusive layer (diffusive gel and filter membrane), $D$ is the diffusion coefficient of the target analyte in the diffusive layer, $t$ is the exposure time, and $A$ is the window area of the DGT device (3.14 cm$^2$).

In the present study, the diffusion coefficients of the target compounds were experimentally determined, and DGT-based compound concentrations in the laboratory and

field tests were determined using Equation (1). DGT devices containing XAD18 binding gel, 0.80 mm thick agarose diffusive gel and PTFE filter membranes were placed into 2.5 L of well-stirred solutions containing the four target compounds (9-FL; 1-CLAQ; Phe, 100 μg L$^{-1}$; 9-NA, 20 μg L$^{-1}$) and 0.01 M NaCl at 25 °C for 12 h. After adsorption was complete, the binding gel was removed from the DGT device and eluted; then, the mass of the compound adsorbed on the binding gel was calculated.

The diffusion coefficients of the four target compounds were also investigated at a lower concentration due to the low levels of PAHs and their derivatives in ambient water. The DGT devices were deployed in 2.5 L of solution containing the target compounds at 3 μg L$^{-1}$, along with 0.01 M NaCl, at a temperature of 25 °C. After deployment for 12 h, the compounds' diffusion coefficients were measured.

The diffusion coefficients of the four target compounds were also investigated at lower concentrations due to the low levels of PAHs and their derivatives in ambient waters. The DGT devices were deployed in 2.5 L of solution containing the target compounds at 3 μg L$^{-1}$, along with 0.01 M NaCl, at a temperature of 25 °C. After deployment for 12 h, the diffusion coefficients of the compounds were measured.

### 2.5. DGT Performance Tests under Different Conditions

### 2.5.1. Effects of pH and IS

To test the effects of IS and pH on DGT performance, the DGT devices were placed for 12 h in two different well-stirred solutions: (a) a 2.5 L solution containing 100 μg L$^{-1}$ 9-FL, 1-CLAQ and Phe and 20 μg L$^{-1}$ 9-NA (IS = 0.01 M NaCl) at different pH values (5–8, adjusted with 0.1 M HCl or 0.1 M NaOH); (b) a 2.5 L solution containing 100 μg L$^{-1}$ 9-FL, 1-CLAQ and Phe and 20 μg L$^{-1}$ 9-NA (pH = 6) with NaCl concentrations ranging from 0.005 to 0.5 M.

### 2.5.2. Influences of Diffusion Film Thickness and Deployment Time

To explore how the mass taken up by the DGT device was affected by the diffusive gel thickness, DGT devices containing agarose diffusive gels of various thicknesses (0.68 mm–2.18 mm) were immersed in 2.5 L of solution containing 100 μg L$^{-1}$ of 9-FL, 1-CLAQ and Phe, 20 μg L$^{-1}$ of 9-NA and 0.01 M NaCl for 12 h.

To investigate the impact of the deployment time, the DGT devices were immersed in well-stirred solutions (4 L) containing 3 μg L$^{-1}$ of the target compound and 0.01 M NaCl for a period of 7 days (168 h), with three devices being recovered every other day. To ensure accurate results and minimize any potential effects from compound evaporation, the solutions were replaced once daily.

### 2.5.3. Competition Effects and Aging Effects

In order to investigate the adsorption competition among the compounds, the DGT devices were deployed for 12 h in four groups of solutions (2.5 L). The compound concentrations in each group were as follows: (a) 9-FL, 1-CLAQ, Phe: 100 μg L$^{-1}$, 9-NA: 20 μg L$^{-1}$, IS: 0.01 M NaCl; (b) 9-NA, 1-CLAQ, Phe: 100 μg L$^{-1}$, 9-FL: 20 μg L$^{-1}$, IS: 0.01 M NaCl; (c) 9-NA, 9-FL, Phe: 100 μg L$^{-1}$, 1-CLAQ: 20 μg L$^{-1}$, IS: 0.01 M NaCl; (d) 9-FL, 9-NA, 1-CLAQ: 100 μg L$^{-1}$, Phe: 20 μg L$^{-1}$, IS: 0.01 M NaCl.

The XAD18 binding gels were immediately stored in ultrapure water after preparation; the overall storage duration is called the aging time. The aging times of XAD18 binding gels were set to 25, 45 and 90 days. The DGT devices assembled with binding gels with different aging times were placed into 4 L solutions containing 100 μg L$^{-1}$ 9-FL, 1-CLAQ and Phe, 20 μg L$^{-1}$ 9-NA and 0.01 M NaCl for 12 h.

In each performance test described above, the mass of compounds present in the DGT binding gel was measured after device retrieval. Subsequently, the $C_{DGT}$ was calculated using Equation (1) and compared to the compound concentrations in the actual water samples.

*2.6. Field Trials*

The field deployments included two phases, one for evaluation and the other for water quality monitoring. First, we deployed the DGT devices in Goose Pond, a landscape water body at Anhui University campus (117°27′12.46″. 31°37′53.82″) in Heifei City, China, and recorded the water temperature every 15 min utilizing a temperature recorder. Three DGT devices were removed after 3, 6, 10 and 15 d, and the concentrations ($C_{DGT}$) of 9-FL, 1-CLAQ, 9-NA, and Phe were calculated according to Equation (1). Meanwhile, active sampling, i.e., traditional grab sampling, was also conducted at the pond on Day 3, 10 and 15 of DGT deployment to measure the concentrations ($C_{active}$) of the same compounds. Then, the two results were compared. After validating the feasibility of using the XAD18-DGT technique to assess the quantity of four target compounds in the campus pond, we deployed the DGT devices in several rivers flowing into Chaohu Lake, the fifth largest freshwater lake in China, located in Hefei City, during June to September 2023. A map of the sampling sites is shown in Figure 1. The DGT units were retrieved after 7 days at every site. The pretreatment of water samples is presented in the SI.

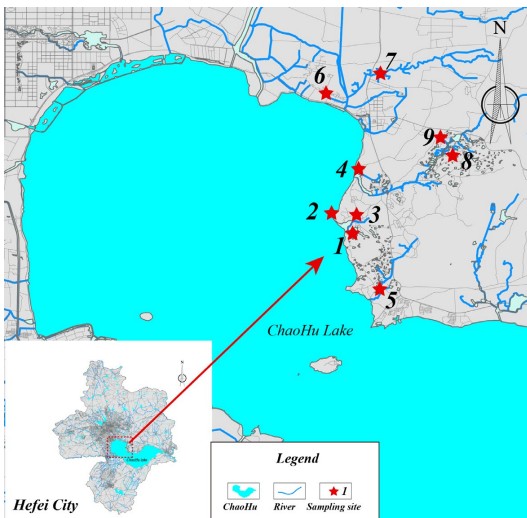

**Figure 1.** DGT device deployment locations: (1) a pond connected to Chibi River, (2) ChaoHu Lake, (3) Chibi River, (4) Lower Yudai River, (5) Heishi River, (6) Changlin River, (7) Changlin River, (8) a pond connected to Yudai River, (9) Yudai River.

## 3. Results and Discussion

*3.1. DGT Device Preparation*

The concentrations of the compounds in ambient water were calculated using the mass of the adsorbed compounds on the binding gel; therefore, the adsorption of compounds on filter membranes and diffusion gels should be avoided as much as possible when the compounds pass them. The test results in Figure 2 demonstrate the low adsorption of the four target compounds onto the diffusion gel (<5%). The DGT molding exhibited a very low adsorption rate for the four target compounds (<10%). Of the four filter membranes, the PTFE membrane exhibited the lowest adsorption rate (5%) for the four compounds, while the other three filter membranes showed much higher adsorption rates (>30%). Therefore, DGT moldings, agarose diffusive gels and PTFE filter membranes were used to construct the DGT devices used in the present study.

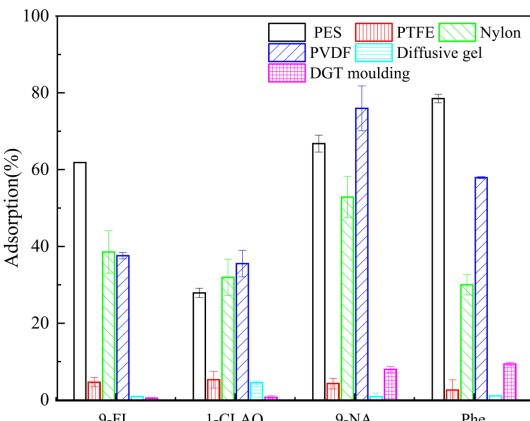

**Figure 2.** Adsorption of 9-FL, 1-CLAQ, 9-NA and Phe onto DGT moldings, diffusive gels, and four different filter membranes (PES, PVDF, PTFE, nylon). Error bars were calculated from the standard deviation of three replicates.

### 3.2. *Characteristics of the Binding Gel*

#### 3.2.1. Adsorption Kinetics

The adsorption of compounds on the XAD18 binding gel increased linearly with time during the first 60 min (Figure 3). The adsorbed mass reached 80% of the total in the solution after 240 min, and the adsorption reached dynamic equilibrium after 720 min. According to Fick's law of diffusion, the amount of compound adsorbed by the binding gel should not be less than the theoretical diffusion amount through the diffusion gel in the first 5 min, ensuring the rapid adsorption of the target compounds; thus, the concentration at the interface between the diffusion gel and binding gel is effectively zero [32]. The average adsorption rates of 9-FL, 9-NA, 1-CLAQ and Phe in the first five minutes were 0.49 ng cm$^{-2}$ s$^{-1}$, 0.13 ng cm$^{-2}$ s$^{-1}$, 0.59 ng cm$^{-2}$ s$^{-1}$ and 0.37 ng cm$^{-2}$ s$^{-1}$, respectively. The compound adsorption rates far exceeded the flux through the DGT diffusion layer (0.80 mm diffusion gel + 0.18 mm filter membrane) (0.00044–0.0048 ng cm$^{-2}$ s$^{-1}$). These results indicate that the XAD18 binding gels could rapidly adsorb the four compounds, effectively maintaining a zero concentration at the interface between the binding and diffusion gels, which is a key criterion for high-accuracy DGT measurements.

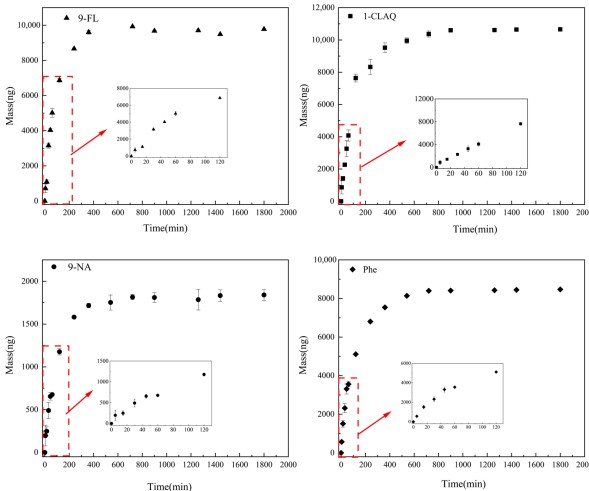

**Figure 3.** The mass of the target compounds accumulated on XAD18 binding gels was measured over time. The solution initially contained 0.01 M NaCl and the target compounds (9-FL; 1-CLAQ; Phe: 100 μg L$^{-1}$; 9-NA: 20 μg L$^{-1}$). Error bars were calculated from the standard deviation of the replicates (*n* = 3).

### 3.2.2. Elution Efficiencies

Ensuring a stable elution efficiency for a specific analyte is crucial for accurately assessing the masses adsorbed by the binding gels. This, in turn, guarantees proper calculation of DGT-based concentrations [14]. XAD18 binding gels, which had adsorbed a known mass of the target compound, were sonicated in 10 mL of methanol for 90 min. The compound elution efficiencies were measured to be within the range of 87.07% to 98.14% (Table 1). This test was also conducted using dichloromethane, but the elution efficiency exceeded 120%, possibly due to the high volatility of this solvent. To ensure accurate measurements, 10 mL of methanol was used for elution in this study.

**Table 1.** Elution efficiencies of target compounds for XAD18 binding gels loaded with 450, 900 and 4500 ng of 9-FL, 1-CLAQ, 9-NA and Phe (%).

| Compound | Mass of the Compound Loaded on the Binding Gels | | | | |
|---|---|---|---|---|---|
| | 450 ng | 900 ng | 4500 ng | Mean | S.D. |
| 9-FL | 94.88 ± 3.08 | 103.21 ± 7.66 | 96.32 ± 1.60 | 98.14 | 4.45 |
| 1-CLAQ | 92.81 ± 4.59 | 90.54 ± 3.31 | 86.16 ± 1.67 | 89.84 | 3.38 |
| 9-NA | 93.08 ± 1.35 | 84.33 ± 4.12 | 83.81 ± 1.29 | 87.07 | 5.21 |
| Phe | 95.18 ± 6.85 | 99.04 ± 0.19 | 96.99 ± 1.35 | 97.07 | 1.93 |

### 3.2.3. Adsorption Isotherm

The results of isotherm tests were fitted to the Langmuir and Freundlich equations (SI Table S3). The correlation coefficients obtained from the Langmuir fit ($R^2$) were smaller than those obtained from the Freundlich fit, meaning that the adsorption process was not single molecule adsorption. The Freundlich model better-described the adsorption behavior of XAD18 for the four target compounds (SI Figure S2). According to a previous study, the concentrations of 9-FL, 1-CLAQ, 9-NA and Phe in environmental waters do not exceed 1 μg L$^{-1}$ [5,33,34]. When the concentration of each type of target substance in the waters was 1 μg L$^{-1}$, the adsorption capacities of 9-FL, 1-CLAQ, 9-NA and Phe were calculated according to the Freundlich equation as 1.233 μg cm$^{-2}$, 1.660 μg cm$^{-2}$, 0.622 μg cm$^{-2}$ and 0.489 μg cm$^{-2}$, respectively. Based on these results, the XAD18-DGT devices could be deployed in an aquatic environment for a minimum of 5–6 months before reaching saturation, allowing the DGT samplers to be used in the field for days to months.

### 3.3. Diffusion Coefficient

The diffusion coefficients (*D*) calculated using Equation (1) were $5.30 \times 10^{-6}$ cm$^2$ s$^{-1}$, $4.51 \times 10^{-6}$ cm$^2$ s$^{-1}$, $4.03 \times 10^{-6}$ cm$^2$ s$^{-1}$ and $3.34 \times 10^{-6}$ cm$^2$ s$^{-1}$ for 9-FL, 1-CLAQ, 9-NA and Phe at 25 °C (Table 2). The diffusion coefficient is affected by temperature, and the *D* values at other temperatures can be estimated using Equation (2). The diffusion coefficients of various compounds calculated in experiments at 20 °C and 28 °C differed by less than 10% from the values calculated by Equation (2). These results demonstrate the accuracy of applying Equation (2) to deduce the diffusion coefficients at different temperatures [26].

$$\log D_T = \frac{1.37023(T-25) + 8.36 \times 10^{-4}(T-25)^2}{109 + T} + \log \frac{D_{25}(273 + T)}{298} \tag{2}$$

where $D_T$ (cm$^2$ s$^{-1}$) is the analyte diffusion coefficient at temperature *T* (°C), and $D_{25}$ (cm$^2$ s$^{-1}$) is the analyte diffusion coefficient at 25 °C.

**Table 2.** Diffusion coefficient values. Experimentally measured D values ($D_{DGT}$) and D values estimated via Equation (2) ($D_{eq}$) and the ratios ($D_{DGT}/D_{eq}$) of four target compounds at 20 °C, 25 °C and 28 °C.

| T (°C) | | Compound | | | |
|---|---|---|---|---|---|
| | | **9-FL** | **1-CLAQ** | **9-NA** | **Phe** |
| 25 °C | $D_{DGT}$ (cm² s⁻¹) | $5.30 \times 10^{-6}$ | $4.51 \times 10^{-6}$ | $4.03 \times 10$ | $3.34 \times 10^{-6}$ |
| 20 °C | $D_{DGT}$ (cm² s⁻¹) | $4.47 \times 10^{-6}$ | $3.72 \times 10^{-6}$ | $3.36 \times 10^{-6}$ | $2.67 \times 10^{-6}$ |
| | $D_{eq}$ (cm² s⁻¹) | $4.61 \times 10^{-6}$ | $3.93 \times 10^{-6}$ | $3.51 \times 10^{-6}$ | $2.91 \times 10^{-6}$ |
| | $D_{DGT}/D_{eq}$ | 0.97 | 0.95 | 0.96 | 0.92 |
| 28 °C | $D_{DGT}$ (cm² s⁻¹) | $5.68 \times 10^{-6}$ | $4.98 \times 10^{-6}$ | $4.13 \times 10^{-6}$ | $3.74 \times 10^{-6}$ |
| | $D_{eq}$ (cm² s⁻¹) | $5.74 \times 10^{-6}$ | $4.88 \times 10^{-6}$ | $4.36 \times 10^{-6}$ | $3.62 \times 10^{-6}$ |
| | $D_{DGT}/D_{eq}$ | 0.99 | 1.02 | 0.95 | 1.03 |

The diffusion coefficients of 9-FL, 1-CLAQ, 9-NA and Phe determined at a solution concentration of 3 µg L⁻¹ at 25 °C were $5.07 \times 10^{-6}$ cm² s⁻¹, $4.50 \times 10^{-6}$ cm² s⁻¹, $3.79 \times 10^{-6}$ cm² s⁻¹ and $3.68 \times 10^{-6}$ cm² s⁻¹, respectively. The difference between the diffusion coefficient values measured under the two levels of solution concentrations does not exceed 10%. The test results demonstrate that variations in the concentration of target compounds in ambient water have little impact on the diffusion coefficient. This finding confirms the stability and reliability of utilizing DGT devices to evaluate PAH derivatives and PAH pollution in diverse water bodies.

### 3.4. Factors Affecting DGT Performance

#### 3.4.1. Effects of pH and IS

At pH 5–8, the concentrations of the four compounds measured using the DGT ($C_{DGT}$) were very close to the actual concentrations ($C_{soln}$), with $C_{DGT}/C_{soln}$ ranging from 0.9 to 1.1 (Figure 4a), which indicates that the XAD18-DGT device can adapt well to the changes in acidity and alkalinity in water.

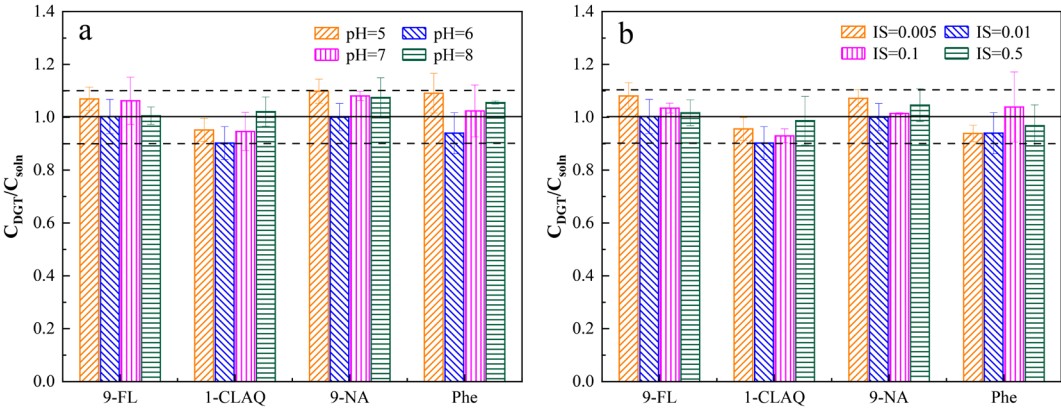

**Figure 4.** The ratio of $C_{DG}T$ to $C_{soln}$ of 9-FL, 1-CLAQ, 9-NA and Phe under different pH values (**a**) and ionic strengths (**b**). The solid horizontal lines represent the target value of 1, and the dotted horizontal lines represent target values of 0.9 and 1.1. Values are means ± SD of three replicate analyses.

Figure 4b shows that the DGT device results for the four compounds were not significantly influenced by the IS of the solutions (0.001–0.5 M of NaCl). The ratios of $C_{DGT}/C_{soln}$ ranged from 0.9 to 1.1, indicating that the XAD18-DGT device may be applied not only fresh in water but also in seawater.

3.4.2. Effects of Diffusion Film Thickness and Deployment Time

Based on Equation (1), in a well-stirred solution with the same analyte concentration and temperature, the adsorbed mass of the target analyte in the DGT devices should exhibit a linear relationship with the reciprocal of the diffusion layer thickness (diffusion gel + filter membrane). As shown in SI Figure S3, the mass of the target compound adsorbed on the binding gel increased linearly ($r^2$ = 0.978–0.995, $p < 0.05$) with the inverse of the diffusion layer thickness (0.68–2.18 mm). The experimental results for different diffusion layer thicknesses were in good accordance with the values calculated via Equation (1), certifying that the mass transfer in the DGT device obeyed Equation (1) and that the diffusive boundary layer thickness does not influence the DGT device's measurement accuracy. The results also verify the accuracy of the diffusion coefficients determined in the present study.

In the aquatic environment, PAH derivative and PAH concentrations are usually at ng L$^{-1}$ levels [5,9]; thus, DGT devices need to be deployed for long periods in order to accumulate the measured compounds at detectable levels. The masses of the four compounds measured by the XAD18-DGT device increased linearly with deployment time (12–168 h) (SI Figure S4) and were in good agreement with the theoretical values predicted by Equation (1). The results show the deployment time (within a certain range) does not influence the DGT device's performance when deployed in ambient water.

3.4.3. Competition of Compounds and Aging Effect of XAD18 Binding Gels

The $C_{DGT}/C_{soln}$ values of the solutions containing the four target compounds with different concentration ratios were in the range of 0.9 to 1.1 (SI Table S4), quite close to the values measured in the single compound solution test, indicating that the DGT devices could be used for simultaneous monitoring of the four target compounds in water. The results show that the DGT measurement was likely not affected by the analyte concentration levels or the presence of other trace organic compounds in environmental waters.

When DGT devices are deployed in ambient water, the DGT plastic molding and filter membrane can protect the binding gel from the effects of the external environment, so membrane aging is an important factor that might affect the performance of the binding gel. The binding gels were prepared and placed in ultrapure water for 25 to 90 days, and then their adsorption properties were measured. The $C_{DGT}/C_{soln}$ values ranged from 0.9 to 1.1 (SI Figure S5), indicating that the binding gel performance was not affected by the aging time set in this study.

*3.5. Field Evaluation and Application*

The XAD18-DGT device was deployed in Goose Pond for evaluation. The concentrations of compounds measured using DGT and active (grab) sampling are shown in Figure 5. The temperature in the pond varied in the range of 28 ± 4 °C, recorded by a temperature recorder at 15 min intervals (SI Table S5). The time-integrated concentrations of 9-FL, 1-CLAQ, 9-NA and Phe measured by DGT sampling were 45.37 ng L$^{-1}$, 22.38 ng L$^{-1}$, 14.68 ng L$^{-1}$ and 87.71 ng L$^{-1}$, respectively, and the compound concentrations determined via grab sampling were 53.13 ng L$^{-1}$, 7.65 ng L$^{-1}$, 4.55 ng L$^{-1}$ and 84.06 ng L$^{-1}$, respectively, as shown in Figure 5. The concentrations of 9-FL and Phe determined by the DGT technique were close to the values determined by active sampling, while the concentrations of 1-CLAQ and 9-NA were very different between DGT and active sampling. The sources of PAH derivatives and PAHs in the campus pond are mainly atmospheric dust fall, rainwater and surface runoff. Therefore, the 9-NA and 1-CLAQ concentrations in the pond are low, and it is thus difficult to extract and concentrate 9-NA and 1-CLAQ from the collected ambient water samples, which may result in large experimental errors in the measurement results. This tedious extraction and concentration process could be avoided by using DGT sampling. The experimental results further verify the advantages and accuracy of using the DGT device for the determination of PAH derivatives and PAHs in waters.

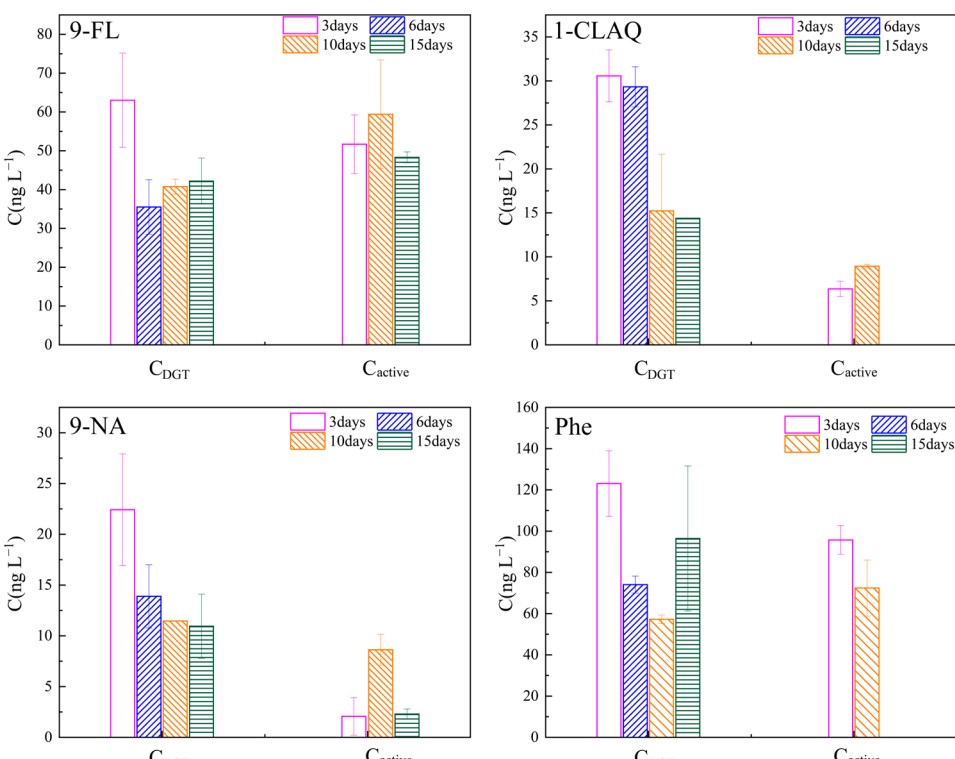

**Figure 5.** Passive sampling results (C$_{DGT}$) and active sampling results (C$_{active}$) for target compounds in Goose Pond. Values are means ± SD of three replicate analyses.

In the 15-day deployment experiments, the results show that retrieval of the DGT devices after 3 days of deployment might result in low accumulation of the target compounds on the binding gel, making the assay difficult. However, retrieval of the DGT devices after 15 days might result in biofilm growth on the DGT filtration membrane layer, which would affect the diffusion rate and lead to inaccurate assay results. Thus, the optimal deployment period for DGT is likely 6–10 days in natural waters.

After evaluating the feasibility of environmental application and the optimal deployment time, 14 groups of XAD18 DGT devices were deployed in the rivers around the Chaohu Lake Basin. However, only nine groups were recovered. The specific deployment sites are shown in Figure 1, and the target compound concentrations at each site are shown in Figure 6. The mean concentrations of Phe, 9-FL, 1-CLAQ and 9-NA at the sampling sites were 147.8–208.2 ng L$^{-1}$, 17.7–48.5 ng L$^{-1}$, 14.9–33.5 ng L$^{-1}$ and 6.6–24.9 ng L$^{-1}$, respectively. According to previous studies of surface waters, the concentrations of 9-FL and 1-CLAQ are about 16–33 ng L$^{-1}$ and 13–20 ng L$^{-1}$, respectively [5,34], the concentration of 9-NA is about 4–14 ng L$^{-1}$ [35], and the concentration of Phe is about 22–103 ng L$^{-1}$ [33,34]. In this study, the concentrations of the four target compounds in rivers, determined via the DGT technique, were slightly higher than those measured in natural waters in the above studies. This might be due to the fact that the monitored sites selected in this study were located in the Chaohu Lake watershed, which has a dense population. The rivers in this watershed are affected by lots of industry and agriculture pollutant input and are relatively seriously polluted.

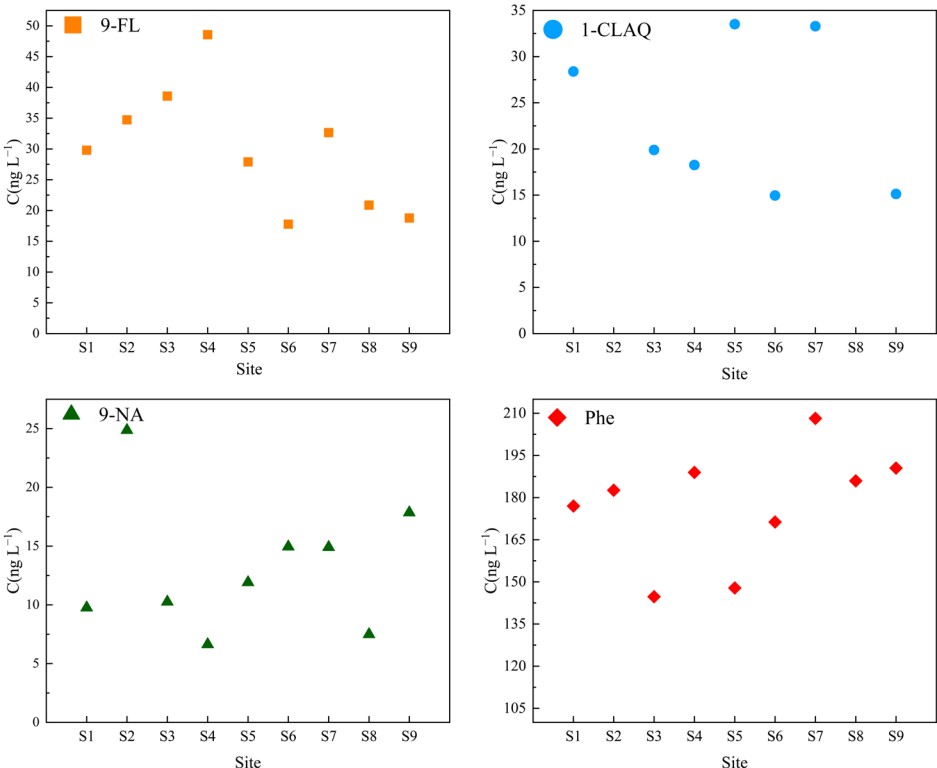

**Figure 6.** Concentrations of 9-FL, 1-CLAQ, 9-NA and Phe in rivers in the Chaohu Lake watershed measured using the DGT technique at nine sites. The deployment site details are illustrated in Figure 1.

## 4. Conclusions

In this work, a feasibility study for using the XAD18-DGT technique for the detection of PAH derivatives and PAHs in waters was conducted, proving that it is a reliable and cost-effective method for PAH derivative and PAH detection. Although only four typical target compounds were studied, this class of organic compounds have similar physicochemical properties, so the DGT technique could be applied to monitor other PAH derivatives and PAHs. The approach was stable and reliable for measuring PAH derivatives and PAHs, and it was not limited by the pH, ionic strength or deployment time within a certain range in natural waters. Thus, DGT devices could be used in a variety of aqueous environments, such as highly polluted fresh water and seawater. The resulting concentrations measured using the DGT technique are time-weighted concentrations, which better reflect the true levels of organic pollutants in water bodies. By employing the DGT technique, the potential risks associated with sampling in adverse weather conditions are mitigated. Compared with other passive samplers, the DGT method's flux independence and ease of use without field calibration could make it a suitable technique for in situ monitoring of PAH derivatives and PAHs in a variety of aquatic systems.

**Supplementary Materials:** The following supporting information can be downloaded at: https://www.mdpi.com/article/10.3390/w16050684/s1, Table S1. Physicochemical properties of 9-FL, 1-CLAQ, 9-NA and Phe; Table S2. Reagents and materials used in the test; Table S3. Parameters of Langmuir and Freundlich isotherms for 9-FL, 1-CLAQ, 9-NA and Phe; Table S4. The ratio of concentrations measured by DGT ($C_{DGT}$) to concentrations in solutions ($C_{soln}$) with different concentration ratios of the four target compounds. Values were expressed as mean $\pm$ standard deviation of at least three replicates; Table S5. Water temperature of Goose Pound, located in Anhui University campus, recorded by temperature loggers with different deployment periods; Table S6. Optimal instrumental parameters of HPLC used in this study; Figure S1. DGT Device Structure Schematic; Figure S2. Adsorption isotherm of 9-FL, 1-CLAQ, 9-NA and Phe for XAD18 binding gels.

Error bars were calculated from at least three replicates; Figure S3. Measured masses of 9-FL, 1-CLAQ, 9-NA and Phe accumulated by DGT devices with different diffusive layer thickness (0.68−2.18mm) deployed in well-stirred solutions at particular concentration. The solid line represents the theoretical values predicted from the known solution concentrations using Equation (1). Error bars were calculated from at least three replicates; Figure S4. Measured masses of 9-FL, 1-CLAQ, 9-NA and Phe accumulated by DGT over a period of 168 h. The DGT devices were deployed in 4 L well-stirred solutions containing 0.01 M NaCl and the four target compounds at 3 $\mu g\,L^{-1}$. The solid line represents the theoretical values predicted from the known solution concentrations using Equation (1). Error bars were calculated from three replicates; Figure S5. The ratios of concentrations measured by the XAD18-DGT ($C_{DGT}$) with different aging time, to the known solution concentrations ($C_{soln}$) of the four target analytes. The solid horizontal line represents the target value of 1, and the dotted horizontal lines represent target values at 0.9 and 1.1. Values are means $\pm$ SD of three replicate analyses; Figure S6. Some pictures of the entire DGT field deployment process. References [26,27] are cited in Supplementary Materials.

**Author Contributions:** S.R.: conceptualization, methodology, investigation, writing—original draft; L.L.: validation, investigation; Y.L.: visualization, funding acquisition; J.W.: validation, writing—reviewing and editing; Y.D.: conceptualization, writing—reviewing and editing, supervision, funding acquisition. All authors have read and agreed to the published version of the manuscript.

**Funding:** This research was financially supported by the agriculture non-point source pollution control project of Feidong county, China (No. 2023ADDFZ00164).

**Data Availability Statement:** Date are contained within the article.

**Conflicts of Interest:** The authors declare no conflicts of interest.

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
