# Peer review of "Development and Field Application of a Diffusive Gradients in Thin-Films Passive Sampler for Monitoring Three Polycyclic Aromatic Hydrocarbon Derivatives and One Polycyclic Aromatic Hydrocarbon in Waters"

_water, doi:10.3390/w16050684_

Round 1
Reviewer 1 Report
Comments and Suggestions for Authors
Review of “Development and filed application of a DGT passive sampler for monitoring three polycylic aromatic hydrocarbons (PAH) derivatives and one PAH in waters” by Ren et al.
Major comments: The paper is a very good methods paper. It requires significant edits of English. Before acceptance for publication the figures in the text must be improved. Figures 2, 3, and 4 are low resolution and nearly unreadable at the scale produced. The authors should also consider moving some of the Supplemental Data figures into the text to aid the readers.
Minor comments:
p. 1, line13. The sentence beginning, “Here, we developed….” This sentence needs to be rewritten as it does not make sense.
p. 1, line 20. Change pollutants to pollutant and concentration to concentrations
p. 2, line48. ”pollution over a period…”
p. 2, line 52. Change to provides
p. 2, line 59. “…Gradients in the thin film…:
p. 2, line 79. “….Inexpensive to produce…”
p. 2, line 87. “A standard DGT device….”
p. 3, line 99. Rewrite the sentence “The concentration program is due….”
p. 4, line 147. Change is to was (two times)
p. 4, line 185. device not devices
p. 4, line 190. water not waters
p. 4, line 196. Delete “the” before day 3
p. 5, line 213. “when the compounds pass through them.”
p. 5, line 220. Subheading. Characteristics of the binding gel
p. 7, line 275. Correct typo on Where
Comments on the Quality of English Language
Comments on some improvements to the English were transmitted to the authors again.
Reviewer 2 Report
Comments and Suggestions for Authors
It is a very interesting paper and there are passive samplers available in the market commercially. It is good the authors assembled their own. My only concern is its practical application when there is no quantification of the amount of water/fluid filtered. We shouldn't express the concentration in field conditions then.
Comments on the Quality of English LanguageLanguage is clear.
